# A genomic snapshot of *Salmonella enterica* serovar Typhi in Colombia

**Paula Diaz Guevara**[1], **Mailis Maes**[2,3], **Duy Pham Thanh**[4,5], **Carolina Duarte**[1], **Edna Catering Rodriguez**[1], **Lucy Angeline Montaño**[1], **Thanh Ho Ngoc Dan**[4], **To Nguyen Thi Nguyen**[4], **Megan E. Carey**[2,3], **Josefina Campos**[6], **Isabel Chinen**[6], **Enrique Perez**[7], **Stephen Baker**[2,3]*

**1** Grupo de Microbiología, Instituto Nacional de Salud, Bogotá, Colombia, **2** University of Cambridge School of Clinical Medicine Department of Medicine, Cambridge Biomedical Campus, Cambridge, United Kingdom, **3** Cambridge Institute of Therapeutic Immunology and Infectious Disease, Level 5 Jeffrey Cheah Biomedical Centre, Cambridge Biomedical Campus, University of Cambridge, Cambridge, United Kingdom, **4** The Hospital for Tropical Diseases, Wellcome Trust Major Overseas Programme, Oxford University Clinical Research Unit, Ho Chi Minh City, Vietnam, **5** Centre for Tropical Medicine and Global Health, University of Oxford, Oxford, United Kingdom, **6** Red Pulsenet Latinoamérica y el Caribe, INEI-ANLIS "Dr Carlos Malbran, Buenos Aires, Argentina, **7** Health Emergencies Department, Pan American Health Organization/World Health Organization, PAHO/WHO, Washington DC, United States of America

☺ These authors contributed equally to this work.
* Sgb47@cam.ac.uk

**Data Availability Statement:** All fastq files are available from the ENA database (accession number PRJEB42858).

## Abstract

Little is known about the genetic diversity of *Salmonella enterica* serovar Typhi (*S.* Typhi) circulating in Latin America. It has been observed that typhoid fever is still endemic in this part of the world; however, a lack of standardized blood culture surveillance across Latin American makes estimating the true disease burden problematic. The Colombian National Health Service established a surveillance system for tracking bacterial pathogens, including *S.* Typhi, in 2006. Here, we characterized 77 representative Colombian *S.* Typhi isolates collected between 1997 and 2018 using pulse field gel electrophoresis (PFGE; the accepted genotyping method in Latin America) and whole genome sequencing (WGS). We found that the main *S.* Typhi clades circulating in Colombia were clades 2.5 and 3.5. Notably, the sequenced *S.* Typhi isolates from Colombia were closely related in a global phylogeny. Consequently, these data suggest that these are endemic clades circulating in Colombia. We found that AMR in *S.* Typhi in Colombia was uncommon, with a small subset of organisms exhibiting mutations associated with reduced susceptibility to fluoroquinolones. This is the first time that *S.* Typhi isolated from Colombia have been characterized by WGS, and after comparing these data with those generated using PFGE, we conclude that PFGE is unsuitable for tracking *S.* Typhi clones and mapping transmission. The genetic diversity of pathogens such as *S.* Typhi is limited in Latin America and should be targeted for future surveillance studies incorporating WGS.

**Funding:** This work was supported by a Wellcome senior research fellowship to SB to (215515/Z/19/Z). DTP is funded as a leadership fellow through the Oak Foundation. Surveillance by the Acute Diarrheal Disease Laboratory was conducted under the Typhoid, Paratyphoid fever and Food Borne Disease Surveillance program as part of the Microbiology Laboratory of the National Health Institute and was supported by a grand from The Administrative Department of Sciences, Technology and Innovation (Colciencias) grand number: 757. Project name: "Fortalecimiento de la capacidad diagnóstica, de investigación y de vigilancia de enfermedades transmisibles emergentes y reemergentes en Colombia". MM is funded by National Institute for Health Research [Cambridge Biomedical Research Centre at the Cambridge University Hospitals NHS Foundation Trust]. PDG received a fellowship from the Enteric infections group at Oxford University Clinical Research Unit, Ho Chi Minh City, Vietnam. The funders had no role in study design, data collection and analysis, decision to publish, or preparation of the manuscript.

**Competing interests:** The authors have declared that no competing interests exist.

## Author summary

*Salmonella* Typhi is the causative agent of typhoid fever, with between 9–13 million cases and 116,800 associated deaths annually. Typhoid fever is still a public health problem mainly in low and middle-income countries (LMICs), including in Latin America, which has a modelled incidence of up to 169 (32–642) cases per 100,000 person-years. Several international studies have aimed to fill data gaps regarding the global distribution and genetic landscape of typhoid; however, in spite of these efforts Latin America is still under-represented. The globally dominant lineages of *S.* Typhi (e.g., H58), which often carry multi-drug resistance (MDR) plasmids, decreased fluoroquinolone susceptibility, and now azithromycin resistance, are not detectable by the accepted method (PFGE) used to track outbreaks of typhoid in Latin America. We compared PFGE with whole genome sequence (WGS) and found it correlated poorly, resulting in the over clustering of cases. We additionally found that unlike in most endemic countries, *S.* Typhi in Colombia are highly antimicrobial susceptible and restricted to a limited number of genotypes that are not as commonly identified in other *S.* Typhi endemic countries. Our study provides the first enhanced insights into the molecular epidemiology of *S.* Typhi in Colombia, using WGS data for the first time to investigate the population structure in Colombia and identifying predominant circulating genotypes. Our work demonstrates that routine surveillance with the integration of WGS is necessary not only to improve disease burden estimates, but also to track the national and regional transmission dynamics of *S.* Typhi.

## Introduction

*Salmonella enterica* serovar Typhi (*S.* Typhi) is the bacterial agent of typhoid fever. With between 9–13 million cases and 116,800 associated deaths annually, typhoid is still a public health problem in many low and middle-income countries (LMICs), particularly in South Asia and parts of sub-Saharan Africa[1,2]. Antimicrobial resistance (AMR) is a major issue, with multi-drug resistance (MDR; resistance to chloramphenicol, ampicillin, and trimethoprim-sulfamethoxazole) and fluoroquinolone resistance in genotype 4.3.1 (H58) organisms dominating the global genetic landscape [3,4]. The emergence of extensively-drug resistant (XDR; MDR and resistant to fluoroquinolones and third generation cephalosporins) in Pakistan and more recent reports of resistance to azithromycin in South Asia compound the problem [5,6]

Several international studies have aimed to fill data gaps regarding the global distribution of typhoid [7–10]. However, there have not been large multicenter population-based surveillance studies conducted in Latin America as there have been in sub-Saharan Africa and South Asia, nor is there routine blood culture surveillance, so this region represents a major data gap in global disease burden estimations [11–13]. The modelled incidence of typhoid in Latin America varies enormously, and estimates range from 1.0 (0.2–3.9) cases and 169 (32–642) cases per 100,000 person-years [8,14]. A lack of systematic surveillance also means that there are limited contemporary data on the circulating bacterial population, AMR profiles, and potential transmission dynamics within South America. However, a recent study revealed a large number of *S.* Typhi isolates with a high prevalence of decreased fluoroquinolone susceptibility in Colombia and El Salvador [15].

Pulsed Field Gel Electrophoresis (PFGE) is the conventional method for studying the genetic relationship between *S.* Typhi isolates in Latin America [16]. Using this method, we recently found that some *S.* Typhi isolates from Colombia shared indistinguishable PFGE patterns with organisms from Argentina, Chile, Perú, Venezuela, Brazil, and Guatemala,

indicative of the circulation of common "continental" genotypes [17,18]. However, PFGE has limited discriminatory power to support subtyping and cannot identify genotype 4.3.1, other emerging genotypes, or AMR genes. Whole genome sequencing (WGS) is the gold standard for the investigating population structures, transmission dynamics, and molecular mechanisms of AMR in *S*. Typhi. In 2015, a landmark global *S*. Typhi genotyping scheme was published, but comprised only 20 genome sequences originating from Latin America (Argentina, El Salvador, French Guiana, and Peru)[3]. Since then, there have been only three additional publications describing *S*. Typhi isolated in Latin America and characterized using WGS, generating a further 36 genome sequences [4,19,20]

In response to local concern regarding the increase of typhoid fever and the international threat of AMR, the National Surveillance System Public Health (SIVIGILA) of Colombia made typhoid fever a notifiable disease in 2006, requiring laboratory follow-up [21]. Here, we aimed to generate the first insights into the molecular epidemiology of typhoid in Colombia by performing AMR profiling and comparative genotyping using both PFGE and WGS on a cross-sectional collection of *S*. Typhi isolated in Colombia between 1997 and 2018.

## Methods

### Ethics statement

This study was conducted in accordance with the principles expressed in the Declaration of Helsinki. The clinical bacterial isolates were collected through the Colombian Laboratory National Surveillance System under the scientific, technical and administrative standard for health research established in Colombian resolution 8430 of 1993 of the Ministry of Health. Patient data were analysed anonymously; consequently, formal ethical approval for the study was not necessary.

### *Salmonella* Typhi isolates

A total of 1,478 *S*. Typhi isolates were submitted to the surveillance program at the National Health Institute of Colombia between 1997 and 2018. These organisms were all associated with a reported typhoid and paratyphoid fever event and came from 22 of 32 Colombian departments and the Capital District of Colombia [21]. 1,077 (72.9%) of these isolates were successfully genotyped using the standard routine PFGE pipeline (S1 Fig) and 77 (5.2%) *S*. Typhi isolates were selected cross-sectionally for WGS (Fig 1 and S1 Table). Our aim was to generate a broad overview of circulating genotypes in Colombia and to identify genotype H58. Therefore, we included isolates from all years, from sampled departments, and a broad range of PFGE patterns, including at least one isolate of each mayor PFGE pattern and including the various AMR phenotypes. These isolates were both from outbreaks defined by the health authorities (n = 12) and sporadic cases (n = 65); 61 isolates originated from blood, 10 from stool, and six from other sources (3 bone marrow, 1 splenic abscess, 1 gluteus abscess, and 1 from a skin swab).

### Bacterial identification and antimicrobial susceptibility testing

All isolates were identified using standard biochemical tests (Triple Sugar Iron Agar (TSI), Citrate, Urea, motility), the automated MicroScan, VITEK II system and the Kauffmann-White-Le Minor scheme to identify organisms suspected to be *S*. Typhi (Difco, United States) [22]

Antimicrobial susceptibility testing was performed using the Kirby-Bauer disk diffusion method against amoxicillin-clavulanic acid (AMC), chloramphenicol (CHL), nalidixic acid (NAL), tetracycline (TET), ampicillin (AMP), cefotaxime (CTX), ceftazidime (CAZ),

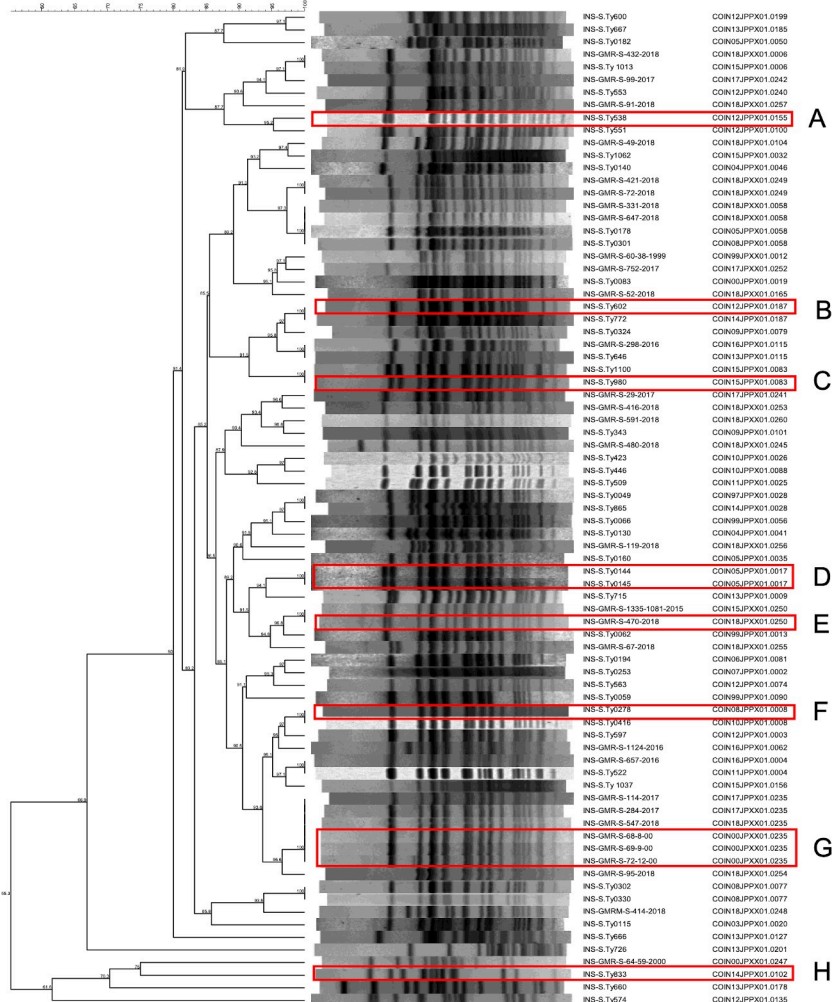

**Fig 1. Colombian *Salmonella* Typhi isolates selected for whole genome sequencing.** PFGE-*XbaI* dendrogram generated with Dice coefficient and UPGMA clustering method (tolerance and optimization 1.5%) of the 77 selected *S.* Typhi isolates with isolate identification and PFGE pattern code. The red boxes indicate isolates from epidemiological confirmed outbreaks (A-H).

trimethoprim-sulfamethoxazole (SXT), and Meropenem in combination with the MIC-based methods using the MicroScansystems according to manufactures recommendations. Ciprofloxacin (CIP) susceptibility was determined by agar dilution assays according to the CLSI standards of 2019 [23] Extended-Spectrum Beta-Lactamase (ESBLs) activity mediated by $bla_{SHV}$, $bla_{TEM}$, and $bla_{CTX-M}$ genes was confirmed by PCR amplification [24].

## Molecular subtyping by PFGE

All organisms were subtyped by PFGE following standardized PulseNet protocols. [16]. Briefly, genomic DNA were digested with *Xba*I (Promega, USA) and subjected to gel electrophoresis. PFGE patterns from the different runs were normalized by aligning the reference digestion pattern of *S.* Braenderup H9812. Bands were assessed visually and by a computerized program (Gelcompare 4.0 software (Applied Maths, Belgium). Parameters of tolerance and optimization were set to 1.5% and similarities calculated according to Dice coefficient. The Clustering dendrogram was based on the unweighted pair-group method using arithmetic

averages (UPGMA). The resulting *Xba*I patterns were compared with the local database and if indistinguishable (within this 1.5% tolerance) from an existing pattern the isolates was given the same PFGE code; if a unique pattern was deteremined a new PFGE code was assigned. All PFGE pattern codes were assigned following the PulseNet International guidelines for nomenclature, which includes 2 letters for the country or region, 3 letters for the serovar, 3 characters for the enzyme and 4 digits for the profile number (e.g. COINJPPX01.0001 for Colombia) [17]

### Genome sequencing and SNP analysis

DNA was extracted using a Qiacube in combination with the Qiagen QIAamp DNA Mini Kit (Qiagen) at the Colombian National Health Institute (INS), following the manufacturer guidelines. DNA was quantified using a Qubit 2.0 fluorometer (Invitrogen) and 2μg of genomic DNA was subjected to indexed WGS by Illumina MiSeq platform to generate 100 bp paired end reads and 30x genome coverage. Genomic libraries were prepared with Nextera XT library prep Kit FC 121–1031. Raw Illumina reads were assembled using (Velvet v1.2) via an automated pipeline at the Wellcome Sanger Institute[25]. For preliminary analysis and global contextualization and for the detection of non-synonymous mutations in the Quinolone Resistance Determining Region (QRDR) of genes *gyrA*, *gyrB*, *parC*, and *are*, the assembled genomes were uploaded to PathogenWatch v3.2.2 (https://pathogen.watch/). Genotypes were assigned using GenoTyphi (https://github.com/katholt/genotyphi)

Sequenced reads and publicly available sequences were mapped and SNP called against the reference genome *S.* Typhi CT18 using the Sanger institute pipelines and following quality metrics as previously described [26]. Known recombinant regions such as prophage[4], were manually excluded, and any remaining recombinant regions were filtered using Gubbins (v1.4.10)[27]. The resultant core SNP alignment of 40,998 bp was used to infer Maximum Likelihood (ML) phylogenies using RAxML (v8.2.8)[28], specifying a generalized time-reversible model and a Gamma distribution to model site-specific rate variation (GTR+ Γ substitution model; GTRGAMMA in RAxML) with 100 bootstrap pseudoreplicates used to assess branch support. SNP distances for the core genome alignment of all the novel genome sequences were calculated from this alignment using snp-dists package (https://github.com/tseemann/snp-dists). SRST2 v0.2.0 [29] was used with the ARGannot [30] and PlasmidFinder [31] databases to detect the molecular determinants associated with AMR; standard cut-offs of >90% gene coverage and a minimum read dept of 5 were used. Maps drawn in inkscape v1.0.1 an open source scalable graphics editor.

## Results

### PFGE genotyping and isolate selection

PFGE is performed routinely for *S.* Typhi in Latin America; results are consolidated into the PulseNet Latin America and Caribbean Network database [16,17]. Organisms are given a unique PFGE code according to their genomic digestion pattern; 1,478 Colombian isolates were present in the national surveillance database at the initiation of this project. We selected 77 *S.* Typhi isolated between 1997 and 2018 to represent the broadest possible diversity (by PFGE; S1 Fig) for WGS. This collection comprised 60 unique PFGE profiles (Fig 1 and S1 Table), including the most commonly circulating restriction patterns in Colombia (e.g., COINXX.JPPX01. 0008-0083-0115)[18]. Twelve isolates also originated from eight outbreaks confirmed by the health authorities (A-H; 8, 4, 24, 9, 5, 2, 6, and 8 patients per outbreak respectively) (Figs 1 and S1); more than one isolate were included from two of these outbreaks (D and G). The selection was skewed towards more recent years based on number of available isolates and for AMR isolates [32].

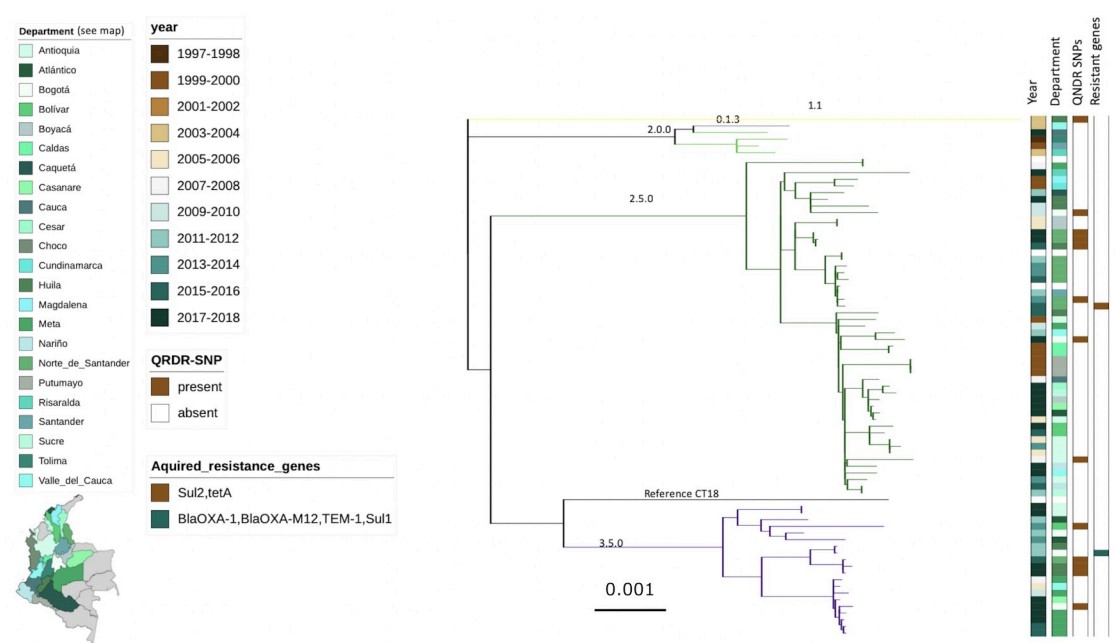

**Fig 2. The phylogenetic structure of *Salmonella* Typhi in Colombia.** SNP based RAxML generated Maximum Likelihood phylogenetic tree of 77 selected *Salmonella* Typhi isolates. Branches are coloured by genotype (numbers shown). Column 1 indicates year of isolation, column 2 indicates department (state) of isolation, column 3 indicates the presence (brown) of SNPs in the QNDR and column D indicates acquired AMR genes (see legend). The map was drawn by inkscape v1.0.1 an Open Source Scalable Vector Graphics Editor.

## AMR and population structure of Colombian *S.* Typhi

The 77 Colombian *S.* Typhi isolates were subjected to WGS and a phylogenetic tree was constructed from core genome SNPs (Fig 2). We found that genotypic variation in the population of Colombian *S.* Typhi was generally limited, with the majority of isolates restricted to two groups: major cluster 2 and 3. These clades could be further segregated into clades 2.5 (51/77; 66.2%), 3.5 (20/77; 24.9%), and 2 (4/77; 5.2%). In addition, we identified two isolates in major cluster 1; these organisms belonged to genotypes 0.1.3 and 1.1.

Notably, unlike a recent observation from Chile, we did not identify genotype 4.3.1 (H58) isolates in this set of Colombian sequences, despite being specifically enriched for organisms that exhibited resistance to antimicrobials. However, we did identify 14 organisms in genotypes 1.1, 2.5 and 3.5 that contained a single SNP in the QRDR region (Fig 2 and Table 1), resulting in reduced susceptibility to fluoroquinolones. Overall, and unlike contemporaneous *S.* Typhi collections from Africa and Asia, this collection contained a limited accumulation of acquired AMR genes. We identified one isolate carrying the *sul*2 and *tet*A genes associated with resistance to tetracycline and sulphonamides, respectively. We additionally detected one organism from Bogota, isolated in 2012, which carried $bla_{CTX-M-12}$, $bla_{TEM-1}$, $bla_{OXA-15}$, and *Sul*1, rendering it resistant to ampicillin, cephalosporins, and sulphamethoxazole (Fig 2 and Table 1).

## Associations between PFGE and WGS

We next aimed to compare the PFGE patterns of the 77 Colombian *S.* Typhi with that of phylogenetic structure created by WGS. First, we found that the paired isolates from the outbreaks (D and G) were indistinguishable; these organisms had identical PFGEs patterns and displayed

**Table 1. Colombian *Salmonella* Typhi isolates with antimicrobial resistance phenotypes and their associated genes and plasmids.**

| Organism id | Genotype | Resistance Kirby-Bauer (mm)* | MIC | Resistance profile | AMR Genes/QRDR mutations | Plasmid content |
|---|---|---|---|---|---|---|
| INS-S.Ty1062/1087-15 | 3.5 | NAL(28),AMP(33), S(14) | AMP >16 CIP 0.008 | AMP | | |
| INS-S.Ty0330/256-08 | 2.5 | NAL(27), AMP(16),S(12) | CIP 0.008 | AMP(I) | | |
| INS-S.Ty0115/36-03 | 2.0 | NAL(28),SXT(6), TET(6), S (14) | CIP 0.008 | SXT-TET | | |
| INS-S.Ty551/52-12 | 3.5 | NAL(24),AMP(6), CTX(8), TET(6), S(6) | CIP 0.008 | AMP-CTX | *bla*CTX-M-12, *bla*OXA-15, *Sul*, *bla*TEM-*1* | IncL/M(pOXA-48) IncFIB (pHCM2) |
| INS-S.Ty980/524-15 | 2.5 | NAL(22), TET(6), S(6) CTX(6), CTX-CLA(10) CAZ(17), CAZ-CLA(19) | AMP >16 CTX >32 CIP 0.016 | AMP-CTX-TET-S | *TetA*(A), *Sul2* | ColRNAI |
| INS-S.Ty0130/69-04 | 1.1 | NAL(6), S(14) | CIP 0.25 | NAL-CIP(I) | *gyrA_S83F* | |
| INS-GMR-S-331-18 | 3.5 | NAL(6),CIP(26)(I) | | NAL-CIP(I) | *gyrA_D87Y* | |
| INS-S.Ty1100/1425-15 | 3.5 | NAL(18)(I) | CIP 0.25 | NAL(I)-CIP(I) | *gyrA_S83F* | |
| INS-S.Ty772/314-14 | 2.5 | NAL(10), CIP(25), S(10) | CIP 0.032 | NAL | *gyrA_D87V* | |
| INS-S.Ty423/316-10 | 2.5 | NAL(6), S(14) | CIP 0.064 | NAL | *gyrA_S83Y* | |
| INS-S.Ty666/329-13 | 3.5 | NAL(6), S(11) | CIP 0.032 | NAL | *gyrA_D87G* | |
| INS-S.Ty1013/755-15 | 2.5 | NAL(6), S(13) | CIP 0.064 | NAL | *gyrA_D87G* | |
| INS-GMR-S-752-17 | 2.5 | NAL(6), CIP(25), S(12) | CIP 0.064 | NAL | *gyrA_D87G* | |
| INS-GMR-S-91-18 | 3.5 | NAL(6), CIP(24), S(14) | CIP 0.064 | NAL | *gyrA_S83F* | |
| INS-GMR-S-480-18 | 2.5 | NAL(6), CIP(26), S(13) | CIP 0.064 | NAL | *gyrA_D87G* | |
| INS-GMR-S-432-18 | 3.5 | NAL(6), CIP(23) | ND | NAL | *gyrA_S83F* | |
| INS-S.Ty0253/397-07 | 2.5 | NAL(6), S(12) | CIP 0.064 | NAL | *gyrA_D87N* | |
| INS-GMR-S-114-17 | 2.5 | NAL(15),CIP(26), S(13) | CIP 0.064 | NAL | *gyrB_S464Y* | |
| INS-S.Ty563/214-12 | 2.5 | NAL(18), CIP(28) | CIP 0.032 | NAL(I) | *gyrA_S83F* | |
| INS-S.Ty538/7-12 | 3.5 | NAL(24), CIP(29), S(12) | CIP 0.008 | Susceptible | | IncFIB(pHCM2) |

*Interpretation criteria according CLSI 2020

no SNP differences in the WGS data (Figs 1 and 3). However, more generally, the PFGE restriction patterns and position in the dendrogram showed minimal concordance with their corresponding phylogenetic location from the WGS data (Fig 3). For example, three isolates from an outbreak (G) shared an identical PFGE restriction pattern (COINXX.JPPX01.0235). This association was encouraging, but on further investigation, an additional three *S*. Typhi isolates exhibited this same restriction profile. These three further isolates had no apparent epidemiological association with the specific outbreak, were from different geographical locations across Colombia, and were isolated several years after the outbreak (Fig 3). These isolates were determined to be >40 SNPs away in the phylogenetic tree from the isolates causing the

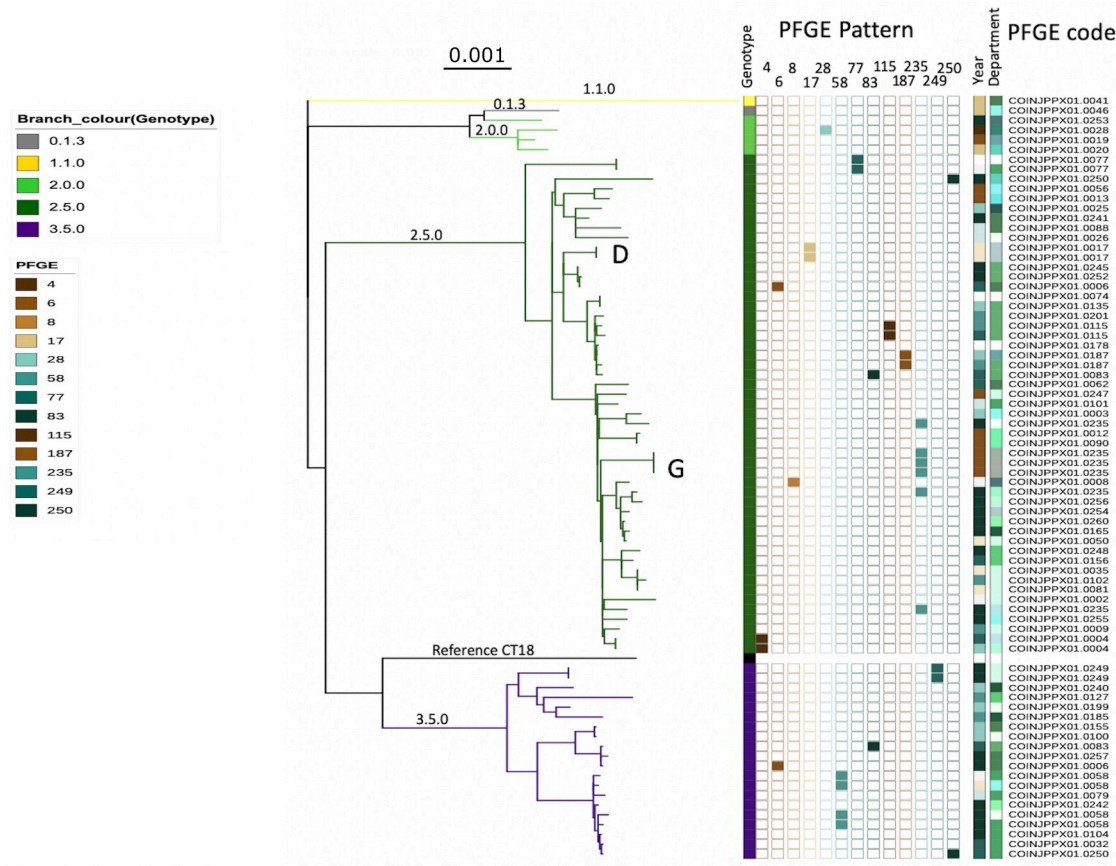

**Fig 3. WGS and PFGE exhibit a limited correlation for *Salmonella* Typhi genotyping.** The branches of the SNP based RAxML phylogenetic tree and the first column are coloured by genotype. The sequential columns highlight the PFGE patterns that were present more than one occasion. i.e., the bottom two isolates of clade 2.5 with PFGE code COINXX.JPPX01.0004 and the top two isolates of clade 3.5 with PFGE code COINXX.JPPX01.0249. All PFGE patterns are additionally listed under PFGE code (the PFGE pattern correspond to the last 4 digits of the PFGE code) for enhanced visibility only unique PFGE are listed by code and not highlighted. The year of isolation and department are coloured as in Fig 2. Letter D and G indicate the two outbreaks from which we sequenced more than one isolate, these correspond with outbreak strains D and G in Fig 1.

outbreak. Lastly, we found a number of occasions where isolates within differing major WGS clades shared an identical PFGE digestion pattern. For example, isolates exhibiting the 0006, 0083, and 0250 PFGE patterns could be found in both clade 2.5 and clade 3.5 of the WGS-based phylogeny (Fig 3). As predicted, these data show that PGFE has limited discriminatory power to identify organisms that may or may not be closely genetically related, further supporting the transition to WGS for routine surveillance.

## Colombian *S.* Typhi in a global context

To determine if the detected Colombian genotypes were more likely to be of Colombian origin or introduced from other continents, we placed these contemporaneous Colombian isolates into a global context with an international collection of *S.* Typhi genome sequences. We constructed a phylogenetic tree of 3,382 publicly available *S.* Typhi genome sequences with the 77 contemporaneous Colombian isolates; genotype 4.3.1 (H58) sequences were excluded as they were not identified in this collection (Fig 4). The Colombian organisms in clades 2.5 and 3.5 clustered alongside other Colombian organisms within their respective genotypes. The nearest

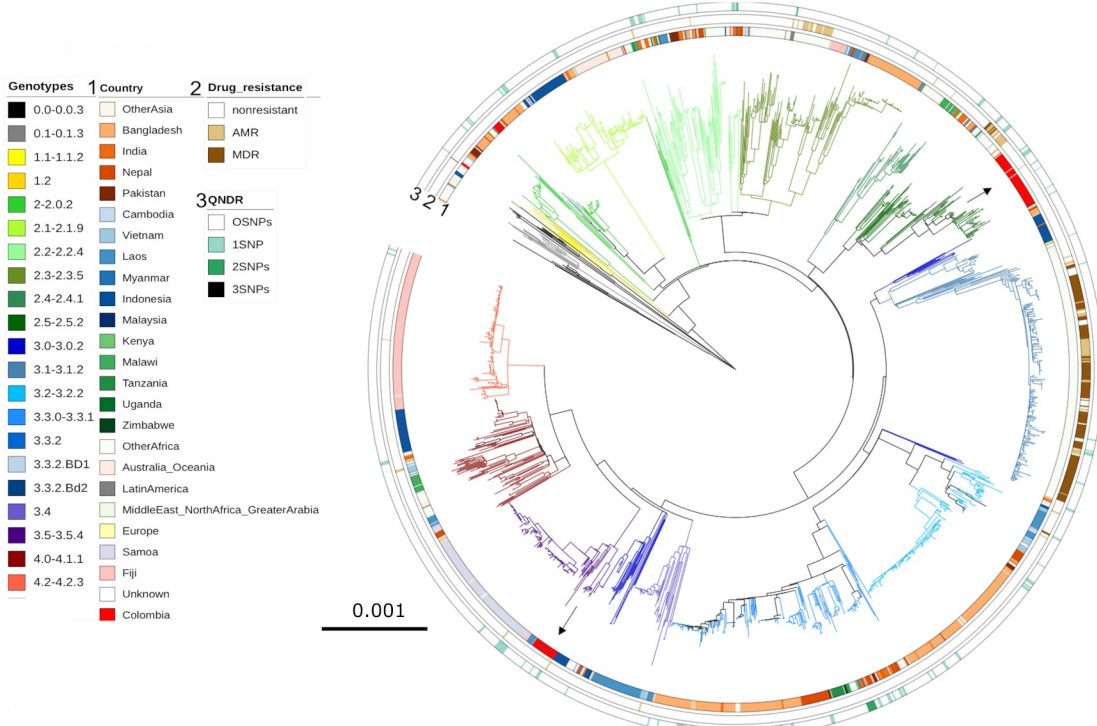

**Fig 4. The phylogenetic location of Colombian *Salmonella* Typhi in a global context.** SNP based RAxML based phylogenetic tree of the 77 Colombian isolates among 3,382 publicly available non-H58 *S.* Typhi genome sequences. Branches are coloured by genotype, inner ring depicts country of isolation, Asia in red shades, South East Asia in blue shades, Africa in green shades, Colombia in bright red and other Latin American countries in grey. The middle ring indicates AMR profile, any AMR is shown in light brown and MDR isolates are shown in dark brown. Finally, the outer ring shows number of QRDR SNPs with the colour intensity increases with increasing number of SNPs.

neighbours to these organisms were isolated in India (10592_2_45, genotype 2.5) and Vietnam (10425_1_60, genotype 3.5) in 1997 and 1993, respectively. In the absence of further sampling, these data suggest that clade 2.5 and clade 3.5 are locally circulating genotypes in Colombia. Similarly, the presence of genotypes 1.1 and 0.1.3 in Colombia is indicative of limited circulation of overseas genotypes. Organisms belonging to genotypes 1.1 and 0.1.3 are considered ancient and presently uncommon on the international *S.* Typhi genotypic landscape and are historically associated with typhoid in Africa [3].

## Discussion

Here, in this primary study of WGS data from *S.* Typhi circulating in Colombia, we show that the majority of organisms in the selected 20-year time span displayed limited genetic diversity, belonging mainly to two major clades: 3.5 and 2.5. The limited number of sequenced isolates from Latin America to date mainly belonged to primary cluster 2 and primary cluster 4, with genotype 2.0 in Argentina, Mexico, El Salvador, and Peru, genotype 2.3.2 in Argentina, El Salvador, and Mexico, genotype 4.1.0 in Brazil [19], and Argentina [3]. In the restricted number of isolates screened here, no genotype 4.3.1 (H58) isolates were found. Although samples were selected to present a maximal variation based on diverse PFGE patterns, included specifically outbreak and AMR related isolates, we cannot be certain H58 is not present in Colombia; however, we can surmise that this genotype is not as broadly distributed as in Asia and Africa. Before 2020, genotype 4.3.1 *S.* Typhi had not been detected in Latin America, but a recent

study identified three independent introductions of H58 into Chile [20]. In Chile, the spread of these isolates appears to have been contained; however, this observation highlights the need for sustained genomic surveillance to detect any additional introductions and potential increased circulation of genotype 4.3.1 [33].

A key observation is that the prevalence of AMR in Colombian *S*. Typhi appears to be significantly lower than that observed in South Asia or Africa. This study, despite being enriched for AMR isolates, indicates an exceptionally low background of AMR in *S*. Typhi, with only one isolate carrying a plasmid containing AMR genes (IncL/M; pOXA-48), with an additional cryptic no-AMR plasmid (IncFIB; pHCM2) also detected. The precise reason(s) for a lower prevalence of AMR in *S*. Typhi, in Colombia are unknown and requires additional investigation. We hypothesise that a lower prevalence of AMR *S*. Typhi, in comparison to Asia and Africa, may be related to antimicrobial access and global pathogen dynamics. Generally, antimicrobial is not better regulated in this region than other locations with a high density of LMICs in the past [34]. However, in the last decade many Latin American countries developed their own National Action Plans to combat AMR under the guidance of PAHO [35]. AMR in *S*. Typhi is not static, and the global trajectory of AMR is increasing; consequently, there is a constant threat of the importation of AMR organisms and sustained surveillance in Colombia remains crucial. These factors highlight the importance of global typhoid surveillance and not purely restricting observations to Africa and Southeast Asia.

We additionally aimed to assess the potential correlations and utility of PFGE for *S*. Typhi tracking across Latin America. We found that PFGE and SNP based phylogenetic do not correlate especially well. We found the same PFGE patterns in completely distinct primary clusters of the SNP based phylogeny. These observations again indicate that PFGE results in false clustering and is not appropriately sensitive for surveillance requiring high resolution delineation of local/regional population structure and dynamics of *S*. Typhi or for outbreak detection in Colombia. WGS is a more appropriate method and is therefore slowly being adopted as the gold standard for these purposes internationally. Lastly, we compared the Colombian isolates to publicly available non-H58 global isolates to determine whether Colombian organisms were imported. This global tree highlighted a lack of genomic information from Latin America. It was therefore impossible to determine whether observed cases are the result of introductions into Colombia from other Latin American countries or local endemic transmission. However, we found that even though the Colombian isolates were collected over 20-years, they formed their own clusters and were not closely related organisms from other locations. These observations suggest that the *S*. Typhi population structure in Colombia is likely driven by sustained endemic circulation of local genotypes.

This study has limitations, the main one being the small sample size of sequenced isolates. The need to select only a subset of samples meant we could have overlooked genotypes and the proportion of the detected genotypes may not be an accurate overview of the distribution. However, this study was aimed to assess *S*. Typhi genetic diversity in Colombia and we show that in spite of our diverse selection of organisms that 90% of the isolates belonged to two predominate clades. More thorough sequencing strategies are required to more accurately determine the distribution of genotypes.

This study provides an enhanced insight into the molecular epidemiology of *S*. Typhi in Colombia, constructing the pathogen population structure and identifying the predominant circulating genotypes. Our work demonstrates that routine surveillance with the integration of WGS is necessary not only to improve disease burden estimates, but also to track the national and regional transmission dynamics of *S*. Typhi and determine AMR profiles. These data will be pivotal to better estimate the burden of typhoid in the region, improve antimicrobial treatment practices and help policymakers to assess the need for typhoid conjugate vaccine

introduction. While the population of *S.* Typhi in Colombia appears isolated, the emergence and spread of AMR variants have been observed internationally [5,6,33]. Consequently, it is critical for improved control and prevention measures that we establish routine WGS surveillance in Colombia and other Latin American countries to strengthen surveillance and monitoring the continental spread of *S.* Typhi.

## Supporting information

**S1 Table. Profile of the organisms selected for sequencing.**
(XLSX)

**S1 Fig. PFGE clustering of Colombian *Salmonella* Typhi isolates.** PFGE-*XbaI* dendrogram generated with Dice coefficient and UPGMA clustering method (tolerance and optimization 1,5%) of 1,077 isolates. The isolates showed 51.45% genetic similarity and represent 211 unique *Xba*I digestion patterns (as of June 2021). The grey dots indicate the isolates selected for WGS.
(PPTX)

## Acknowledgments

We express our thanks to all typhoid fever patients whose isolates were included in this project and the personnel from the local hospitals and public health laboratories in Colombia. We thank the professionals in the Acute Diarrheal Disease Laboratory, specifically those working in the Typhoid, Paratyphoid fever and Food Borne Disease Surveillance program and Microbiology Laboratory of the Colombian National Health Institute. Pulsenet Latin America and Caribbean PNLA&C. Furthermore, we wish to thank Nicholas Thompson for access to the Sanger analysis pipelines and Gordon Dougan for guidance.

## Author Contributions

**Conceptualization:** Paula Diaz Guevara, Stephen Baker.

**Data curation:** Mailis Maes, Megan E. Carey.

**Formal analysis:** Paula Diaz Guevara, Mailis Maes.

**Funding acquisition:** Stephen Baker.

**Investigation:** Paula Diaz Guevara, Mailis Maes, Duy Pham Thanh, Carolina Duarte, Edna Catering Rodriguez, Lucy Angeline Montaño, Thanh Ho Ngoc Dan, To Nguyen Thi Nguyen, Josefina Campos, Isabel Chinen.

**Methodology:** Paula Diaz Guevara, Mailis Maes.

**Project administration:** Stephen Baker.

**Supervision:** Enrique Perez, Stephen Baker.

**Validation:** Mailis Maes.

**Visualization:** Mailis Maes.

**Writing – original draft:** Paula Diaz Guevara, Mailis Maes.

**Writing – review & editing:** Mailis Maes, Duy Pham Thanh, Carolina Duarte, Edna Catering Rodriguez, Lucy Angeline Montaño, Thanh Ho Ngoc Dan, To Nguyen Thi Nguyen, Megan E. Carey, Josefina Campos, Isabel Chinen, Enrique Perez, Stephen Baker.

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
