## [Decision Letter · Decision Letter 0]

8 Jun 2021

Dear Professor Baker,

Thank you very much for submitting your manuscript "A genomic snapshot of Salmonella enterica serovar Typhi in Colombia" for consideration at PLOS Neglected Tropical Diseases. As with all papers reviewed by the journal, your manuscript was reviewed by members of the editorial board and by several independent reviewers. In light of the reviews (below this email), we would like to invite the resubmission of a significantly-revised version that takes into account the reviewers' comments. 

We cannot make any decision about publication until we have seen the revised manuscript and your response to the reviewers' comments. Your revised manuscript is also likely to be sent to reviewers for further evaluation.

Sincerely,

Travis J Bourret

Associate Editor

Sujay Chattopadhyay, PhD

Deputy Editor

Reviewer's Responses to Questions

**Key Review Criteria Required for Acceptance?**

**Methods**

-Are the objectives of the study clearly articulated with a clear testable hypothesis stated?

-Is the study design appropriate to address the stated objectives?

-Is the population clearly described and appropriate for the hypothesis being tested?

-Is the sample size sufficient to ensure adequate power to address the hypothesis being tested?

-Were correct statistical analysis used to support conclusions?

-Are there concerns about ethical or regulatory requirements being met?

Reviewer #1: yes, yes, yes, yes, and yes. No concerns about ethical requirements

Reviewer #2: See below

**Results**

-Does the analysis presented match the analysis plan?

-Are the results clearly and completely presented?

-Are the figures (Tables, Images) of sufficient quality for clarity?

Reviewer #1: Yes analysis and results are clearly and completely presented 

Figures, in my version of the manuscript, are somewhat blurred.

Reviewer #2: See below

**Conclusions**

-Are the conclusions supported by the data presented?

-Are the limitations of analysis clearly described?

-Do the authors discuss how these data can be helpful to advance our understanding of the topic under study?

-Is public health relevance addressed?

Reviewer #1: Yes, yes, yes and yes

Reviewer #2: See below

**Editorial and Data Presentation Modifications?**

Reviewer #1: (No Response)

Reviewer #2: See below

**Summary and General Comments**

Reviewer #1: No weak points in this manuscript. An important paper concerning the molecular epidemiology of S. Thypi. The paper concludes that the S. Typhi population structure in Colombia is mainly driven by endemic circulation of local genotypes. Interesting are the Colombian organisms in clades 2.5 and 3.5. These have their nearest neighbors isolated in India and Vietnam. International trade in agricultural, aquacultural and manufactured food products has facilitated the introduction of new Salmonella serovars within the geographical boundaries of importing countries. Is there perhaps a trade relation between these countries that explains the presence of these genotypes in Colombia? Colombia and Vietnam are the two major coffee export-oriented countries in the world. India and Colombia, as far as I know, have no trade agreements?

Reviewer #2: The manuscript titled “A genomic snapshot of Salmonella enterica serovar Typhi in Colombia” describes the inclusion and analysis of isolates of S. Typhi from Columbia. This manuscript adds new S. Typhi isolates from Colombia, a country on a continent where genome sequences of S. Typhi is largely absent, to the growing global collection of genome data. A technically-sound standardized characterization pipeline was followed using whole genome sequencing, SNP analysis, genotyping, and antimicrobial susceptibility testing. PFGE, in which the study team has extensive expertise, is used as a non-WGS comparator method and determined to be not adequate for the tracking of fine level analysis such as outbreaks and transmission events. While the study makes use of established pipelines and is well done, the primary hypothesis of this work is to address the lack of genomic information from Latin America and Columbia specifically. The data in this study is limited to Columbia, and there is no evidence that would suggest that the trends observed here are generally applicable to all Latin American countries. In fact, the first paragraph of the discussion would suggest that in fact the isolates from Columbia do not represent the rest of the Latin American isolates, even the small subset included there. The manuscript should be rewritten to address Columbia only and not attempt to extrapolate to all of Latin America. The finding that PFGE is not adequate for tracking transmission and outbreaks is not a new finding, but rather one of the reasons many have moved to WGS. This discussion can be removed from the study as it provides very little. Overall, the study is well done but the results and discussion should be reframed to focus on what can be said from this data. Additional comments on focusing the manuscript are included below.

Major comments:

The choice to perform a cross-sectional analysis of 77 (5.2%), rather than a comprehensive analysis of all or a majority of the 1,478 isolates from 1997 to 2018, is not well justified in light of the primary motivations of this manuscript, i.e., to fill in the gap/blind spot of S. Typhi genomics in South America, specifically in Colombia. In a previous publication by the authors [ref 18, doi:10.1371/journal.pntd.0008040], 402 isolates out of 846 reported typhoid cases from 2012-2015 were available for microbiological characterization and genotyping analyzed using PFGE, suggesting many available isolates were either not sequenced or not included. It is unclear why these isolates were not included. Under results (starting with line 184), the authors explain that isolate selections across the 22 years were made to maximize PFGE diversity, but later the authors conclude that PFGE correlated poorly with PFGE. Indeed, WGS of all available isolates could add an interesting investigation into the clonality of the eight presumably distinct outbreaks from which single distinct PFGE isolates were selection (Fig. 1A), if it is fair to assume the outbreaks were designated as such due to a concordance of epidemiological linkage of persons, place, and time with a matching PFGE type. 

In another example where a lack of PFGE and genotype correlation is problematic for the study design, line 211 asserts a lack of genotype 4.3.1 (H58) in Colombia and line 2015 asserts limited accumulation of AMR genes,” but these statements may be false if one or several of the isolates excluded on the basis of PFGE pattern was in fact the H58 clone. Overall, this suggests to me that the inclusion criteria are flawed and should be reconsidered, or the implications of sampling bias should be further discussed. In its present form, the paper is an interim analysis that could be proven wrong by the same isolate collection on which it is based. Ideally, all available isolates would be sequenced and submitted to the public collection via this paper. 

If diverse PFGE patterns is of importance to the authors, a subsequent sub-analysis beginning with a cross-section of the maximal dataset could be suitable in order to reach the conclusion that PFGE does not correlate well with WGS genotype, which is one of the current genotyping standards in the field. Additional methodological detail on how the isolates were determined as the same or different should be included. While many readers will be familiar with this methodology, it is not clear in the current version of the manuscript (i.e. how many bands different were required to determine that two isolates were the same or different).

The authors suggest there are 60 unique PFGE patterns, but only 8 (A-H) are labeled in Figure 1A. The inclusion of the PFGE pattern in Figure 3, where most of the isolates appear to lack these designations is also confusing. Why are there no PFGE designations included. Additionally Figure 1B does not add significantly to the submission and can be removed.

While the PFGE to WGS associations are interesting, they are not surprising considering the difference in the methodology. However, with this dataset it would be good to interrogate the associations between the actual phenotypes and genotypes for AMR.

Minor comments:

Line 74: missing the word “cases” after “13 million”

Lines 161-162 – what quality metrics were used to include or exclude any of the assembled genomes? Any coverage, quality or contig size cutoffs were used in this study.

How were the AMR genes determined to be present or absent. The current manuscript just indicates that they were screened, without any indication of what is considered present or absent.

Fig 2. Many colors for department are indistinguishable and appear to repeat. Consider recoloring, using numbers instead of colors, or adding a supplementary table with each isolate listed followed by key epidemiological and genotypic variables.

Lines 206-9: commas to separate the list are missing, and capitalization of Column is inconsistent.

Line 254: The source (presumably pathogen.watch) and the genome representation (alignment against CT18 or another local reference, or de novo assembly) of these 3,381 publicly available isolates would be helpful to know, as it is important that genomes are processed in the same way – aligned to CT18 or assembled – before being input into RAxML.

Lines 288-299: The suggestion that there is less AMR in many pathogens in Latina America is not supported by references or additional data and seems to be based on conjecture at this point. These paragraph should be referenced appropriately or removed. 

Line 418 – Data availability – it does not appear as though the data is publicly available with the included project ID. The authors must ensure that all data, both raw sequences and assembled data are publicly available prior to resubmission as both are used in the analyses described in this study.

The figure legends are lacking in any detail.

PLOS authors have the option to publish the peer review history of their article (what does this mean?). If published, this will include your full peer review and any attached files.

Reviewer #1: No

Reviewer #2: No
---

## [Decision Letter · Decision Letter 1]

24 Aug 2021

Dear Dr. Baker,

We are pleased to inform you that your manuscript 'A genomic snapshot of Salmonella enterica serovar Typhi in Colombia' has been provisionally accepted for publication in PLOS Neglected Tropical Diseases.

Best regards,

Travis J Bourret

Associate Editor

Sujay Chattopadhyay, PhD

Deputy Editor

Reviewer's Responses to Questions

**Key Review Criteria Required for Acceptance?**

**Methods**

-Are the objectives of the study clearly articulated with a clear testable hypothesis stated?

-Is the study design appropriate to address the stated objectives?

-Is the population clearly described and appropriate for the hypothesis being tested?

-Is the sample size sufficient to ensure adequate power to address the hypothesis being tested?

-Were correct statistical analysis used to support conclusions?

-Are there concerns about ethical or regulatory requirements being met?

Reviewer #1: I approved the earlier version and as I see that no important changes have been made to this version I also approve this version. An important paper in the field.

Reviewer #2: (No Response)

**Results**

-Does the analysis presented match the analysis plan?

-Are the results clearly and completely presented?

-Are the figures (Tables, Images) of sufficient quality for clarity?

Reviewer #1: Yes

Reviewer #2: (No Response)

**Conclusions**

-Are the conclusions supported by the data presented?

-Are the limitations of analysis clearly described?

-Do the authors discuss how these data can be helpful to advance our understanding of the topic under study?

-Is public health relevance addressed?

Reviewer #1: yes

Reviewer #2: (No Response)

**Editorial and Data Presentation Modifications?**

Reviewer #1: (No Response)

Reviewer #2: Minor Grammar issues:

Line 135: “mayor” should be “major”

Line 155: There is an aberrant period after “protocols.”

Line 289: “antimicrobial is” should be “antimicrobials are”

Like 534: “1,5%” should be “1.5%”

**Summary and General Comments**

Reviewer #1: (No Response)

Reviewer #2: Overall, the authors have accepted that these data will apply to Columbia, but not the rest of South America and have re-written this to reflect that suggestion. They have done a good job addressing the comments of the previous reviewers. The key point where I have an issue is with the lack of data release. While they have released the raw reads, which is great, the assemblies, upon which much of the results are based are not in the public domain. It is difficult, if not impossible to completely review the paper completely without this information in the public domain. There have been far too many retractions and loss of data based on the "promises" to release data upon acceptance of the work.

PLOS authors have the option to publish the peer review history of their article (what does this mean?). If published, this will include your full peer review and any attached files.

Reviewer #1: No

Reviewer #2: No

---

## [Editor Report · Acceptance letter]

8 Sep 2021

Dear Professor Baker,

We are delighted to inform you that your manuscript, "A genomic snapshot of Salmonella enterica serovar Typhi in Colombia," has been formally accepted for publication in PLOS Neglected Tropical Diseases.

Best regards,

Shaden Kamhawi

co-Editor-in-Chief

Paul Brindley

co-Editor-in-Chief
